

# Performance analysis of aspect-level sentiment classification task based on different deep learning models

Feifei Cao and Xiaomin Huang

School of Economics, Guangdong Peizheng College, Guangzhou, China

## ABSTRACT

Aspect-level sentiment classification task (ASCT) is a natural language processing task that aims to correctly identify specific aspects and determine their sentiment polarity from a given target sentence. Deep learning models have been proven to be effective in aspect-based sentiment classification tasks, and the mainstream Aspect-level sentiment classification (ASC) models currently constructed generally assume that the training and test datasets are Gaussian distribution (*e.g.*, the same language). Once the data distribution changes, the ASC model must be retrained on the new distribution data to achieve good performance. However, acquiring a large amount of labeled data again typically requires a lot of manpower and money, which seems unlikely, especially for the ASC task, as it requires aspect-level annotation. This article analyzes the performance of sequence-based models, graph-based convolutional neural networks, and pre-training language models on the aspect-level sentiment classification task using two sets of comment datasets in Chinese and English, from four perspectives: classification performance, performance with different aspect numbers, specific case performance, and computational cost. In this article, we design a state-of-the-art ASC-based classification method and conduct a systematic study on eight public standard English and Chinese datasets with various commonly used assessment measures that provide directions for cross-language migration. Finally, we discuss the limitations of the study as well as future research directions.

Corresponding authors
Feifei Cao, caoff16@lzu.edu.cn
Xiaomin Huang, hxmzy315@126.com

## INTRODUCTION

Aspect-level sentiment classification tasks (ASCTs) are mainly studies based on the analysis of people's opinions, evaluations, attitudes or sentiments expressed about a specific aspect of an entity, which may refer to a product, service or event, *etc.* Commonly, ASCT is based on three main levels: chapter level, paragraph level and sentence level (*Collobert et al., 2011*). Currently, the techniques to contend with the overall sentiment analysis of chapters and paragraphs are relatively mature, but the analysis of specific aspects in chapters and paragraphs is facing challenges (*Goldberg, 2016*), Aspect-level sentiment classification (ASC) based on the sentence level can be well extended to the chapter-paragraph level, so this article aims to investigate ASC at the sentence level. This task can be specifically

divided into two categories: the extraction of an idea, entity or aspect in a sentence; the classification of the sentiment polarity of that idea, entity or aspect's affective polarity classification (which can be subdivided into positive, negative and neutral) (*Schouten & Frasincar, 2016*). This article focuses on the study of the second problem.

Early research in ASC focused on machine learning algorithms (*Nazir et al., 2020*), and with the development of neural networks, the RNN-based prototype approach incorporating attention mechanisms is currently a good approach to solve tasks in the NLP domain and has achieved good results in aspect-level sentiment analysis tasks (*Xing et al., 2019*; *Liu, 2012*; *Turney, 2002*). Methods introducing convolutional neural networks (CNN) followed immediately, such as (*Kim, 2014*; *Sabour, Frosst & Hinton, 2017*; *Liu et al., 2020*), who used CNN -based models to integrate the syntactic and lexical structure of sentences and also showed good performance. In addition, the pre-trained language based model BERT has also achieved great success in sentiment analysis (*Xu et al., 2019*; *Sun, Huang & Qiu, 2019*; *Jiang et al., 2019*). Compared with English sentiment analysis, Chinese sentiment analysis faces several challenges as follows: (1) high-quality Chinese datasets are not abundant enough; "ChnSentiCorp" (*Tan & Zhang, 2008*), "IT168TEST" (*Zagibalov & Carroll, 2008*), "Weibo "4 and "CTB" (*Li et al., 2014*) are four popular Chinese datasets used for general sentiment analysis. However, there is no category information of labeled aspect in these datasets; (2) SentiWordNet, a high-quality dictionary resource, is available in English, but the Chinese domain dictionary resource is not high quality and detailed. In addition, there is a lack of subjective and objective dictionaries; (3) compared with the accuracy rate of English ASC, the accuracy rate is obviously low due to the lack of basic work of Chinese ASC (*Zhang, Li & Song, 2019a*; *Bu et al., 2021*). However, existing studies have systematically classified ASC tasks based on traditional machine learning and deep learning methods, as well as their performance analysis on English benchmark datasets (*Zhou et al, 2019*; *Zhang et al., 2022*; *Poria et al., 2020*). Once the data distribution changes, it is necessary to retrain ASC models for the new distributed data, which requires a lot of labor and money, and it is especially important to find some general ASC model or class of models, and this article hopes to fill that related research. In addition, these models are tested on data sets of synonymous languages, and there is no relevant reference standard for data sets of different languages or different distributions, and this article hopes to provide an answer in this regard. We also give the current high-quality Chinese ASC datasets to better serve the researchers interested in Chinese ASC tasks. We also hope that this study can provide a reference for ASC tasks in other languages.

In summary, this article aims to compare existing deep learning models from four aspects: classification accuracy, specific case performance, performance with different aspect numbers, and time complexity. We will analyze whether these models can achieve the SOTA effect on two datasets with different distributions and draw a conclusion In contrast to other review papers (*Zhou et al, 2019*; *Zhang et al., 2022*; *Poria et al., 2020*), we aim to compare the performance of deep learning language models of different architectures on different data distributions. To the best of our knowledge, we present a fair comparison and systematic analysis of the current state-of-the-art deep learning models in ASC tasks from several aspects.Through this study, we hope to find out the problems of different

models in different data distributions, address the problems of the models, to apply the trained models to different languages, *i.e.,* cross-language migration, to improve the generalization ability of different models to different languages, and to overcome the limitations of the training data will become the focus of ASC task. We also hope that this article can provide a comprehensive understanding of cross-language migration research and provide some insight into the focus of future attention

# LITERATURE REVIEW

In natural language processing, the ASC task was initially tackled with traditional machine learning algorithms, which aimed to make judgments based on syntax and semantics (*Liu et al., 2020*; *Dong et al., 2014*; *Jiang et al., 2011*), but the performance of these models was highly dependent on data quality and therefore unstable. During this period, various types of deep learning models emerged, which we mainly categorized into three types: sequence-based (introducing various forms of RNN), graph-based convolutional neural networks, and pre-trained language models. We mainly review several types of deep learning models used for ASC tasks in recent years and the datasets used.

## Sequence-based for ASC

*Li et al. (2018)* proposed an LSTM method based on an attention mechanism, which introduces an attention mechanism to force the model to focus on the important parts of the sentence. This method has been proven to be an effective way to force neural models to pay attention to relevant parts of the sentence. *Chen et al. (2017)* introduced an end-to-end memory network for aspect-level sentiment classification, which uses an attention mechanism with an external memory to capture the importance of each context word relative to a given target aspect. When inferring aspect sentiment polarity, this method explicitly captures the importance of each context word. In this way, the importance and text representation are computed through multiple computational layers, each of which is a neural attention model on an external memory (*Gu et al., 2018*) combined target identification task with sentiment classification task to better establish the connection between aspect and sentiment. In this way, the signal generated in target detection provides clues for polarity classification, and conversely, the predicted polarity provides feedback for target recognition. *Ma et al. (2017)* use BiGRU to construct the hidden layer, introduce attention to input the result of each time point in the hidden layer into a fully connected layer to produce a probability vector. Then, this probability vector is used to weight each hidden layer result and add them up to obtain a result vector. The introduction of attention can better highlight the importance of sentiment words. PBAN model proposed by *Wang, Huang & Zhao (2016)* not only introduces the attention mechanism but also considers the positional information between sentiment words. PBAN is based on bidirectional GRU. not only pays attention to the positional information of sentiment words but also models the relationship between sentiment words and sentences using bidirectional attention mechanism. *Tay, Tuan & Hui (2017)* combines LSTM and CNN takes into account both the specific aspect information in the sentence and uses location information

## GCN-based for ASC

In addition, deep learning models used for aspect-level sentiment classification tasks mainly adopt graph-based models to integrate syntactic structure. The basic idea is to convert dependency trees into graphs and then link them with graph convolutional networks (GCN) propagate information from syntactic neighbors to aspect words. *Chen, Teng & Zhang (2020)* employs a latent graph structure to complement syntactic features. *Li et al. (2021)* proposed a bipartite graphical convolutional network model, which considers the complementarity of both syntactic structure and semantic relations. *Zhang, Li & Song (2019b)* proposed to build graph convolutional networks on sentence dependency trees to exploit syntactic information and word dependencies. *Bian et al. (2020)* proposed BiGCN, which not only considers the syntactic graph but also considers the lexical graph for capturing dependency relationships in the sentence, taking both the global vocabulary graph and word sequence as input to obtain the initial sentence representation and introducing the HiarAgg module to obtain aspect-oriented representations. *Tian, Chen & Song (2021)* modeled the structure of the sentence through its dependency tree using GCN and also utilized positional information.

## PLMs-based for ASC

Recently, pre-trained language models have also shown excellent performance in aspect-level sentiment classification tasks. *Xu et al. (2019)* proposed a new post-training method based on the BERT model to improve the classification performance of the fine-tuned BERT model on the ASC task.

*Qiu et al. (2020)* proposed a variant of BERT called CG-BERT, which can learn different attention across contexts. The main idea is to first generate a context-aware softmax attention mechanism using a context-aware transformer, and then propose an advanced version of the quasi-attention CG-BERT model, which can learn more important attention components in the context. *Wu et al. (2020)* view ASC as a pairwise sentence classification problem, distinguish Sentiment Polarity by Constructing Sentence Pairs, This can take full advantage of the sentence pair function of the BERT model. *Dai et al. (2021)* fine-tuned RoBERTa by comparing the syntactic tree and dependency parsing tree in pre-trained language models, and introduced an induced tree structure, finding that it outperformed other deep learning models that incorporate dependency parsing trees.

Overall, we counted all the specific methods of these three types of models as shown in Fig. 1 and Table 1. The approaches based on the RNN architecture all use the attention mechanism and all consider aspect-specific word embedding information so that the relevant parts of the sentence can be well captured and the correct sentiment polarity can be obtained; sequence-level classification and sequence-to-sequence model paradigms are used in order to handle specific input and output formats. GCN-based approaches generally take into account sentence-specific aspectual and positional information, aiming at locating each aspect term and better capturing the different aspects and the interaction information before and after the sentence by introducing dependency-tree. The PLMs architecture-based approaches all use dot-product attention and all consider the position information of each word in the sentence to better capture the sentiment expression. These

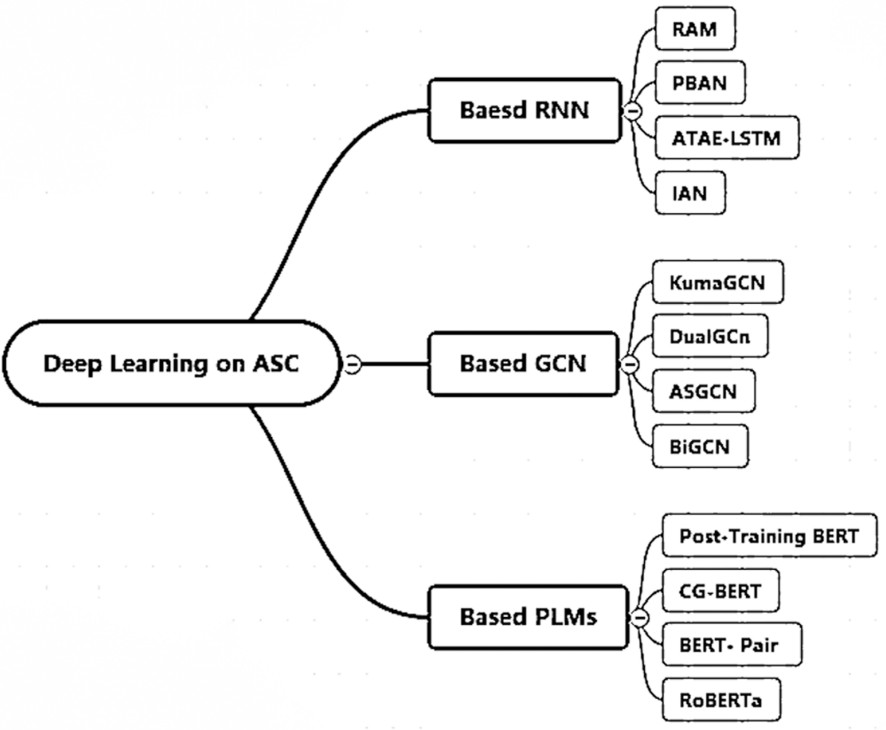

**Figure 1 Deep learning models for ASC task.**

models are very representative of ASC tasks and achieve SOTA' results on their respective benchmark datasets. This is the basis for using them as benchmark models for cross-lingual sentiment polarity analysis in this article.

## Dataset overview

This section provides an overview of sentiment classification datasets commonly used in different deep model architectures in the ASC literature, including information on their language, annotated sentiment elements, *etc*. The most used benchmark datasets in the literature are SemEval-2014 (*Pontiki et al., 2014*), SemEval-2015 (*Pontiki et al., 2015*), and SemEval-2016 (*Pontiki et al., 2016*), which are derived from user reviews of notebooks and restaurants and contain aspect categories and sentiment polarity, all at the sentence level, and thus can be directly used for ASC tasks. In addition, ASC-QA (*Wang et al., 2019a*), ASAP (*Bu et al., 2021*), ACOS (*Cai, Xia & Yu, 2021*), ABSA-QUAD (*Zhang et al., 2021*), Twitter (*Dong et al., 2014*), Sentihood (*Saeidi et al., 2016*), and Mitchell (*Zhang, Zhang & Vo, 2016*) have also been used for ASC. The overall picture of these datasets is given by Table 2, which lists the published studies using these datasets, from which it can be noticed that most of the studies are based on English datasets, with few studies on ASC in other languages.

Can these deep learning models based on aspect-level sentiment classification task achieve the same classification performance on Chinese datasets? Four English datasets and

**Table 1 Statistics of the three types of ASC methods.** Model represents the type of deep learning methods adopted by the corresponding published paper. Aspect and position with a check mark (✓) denotes the model considering the aspect information and position information, respectively. Attention without X means the model using the attention, CA, GA and DPA indicate "Contact Attention", "General Attention" and "Dot-Product Attention" respectively. Class-level indicate different modeling paradigms, SeqClass, Seq2Seq, DepenTree and TokenClass indicate "Sequence-level Classification", "Sequence-to-Sequence", "Dependency-Tree" and "Token-level Classification", respectively.

| | Model | Attention | Aspect | Position | Modeling paradigms |
|---|---|---|---|---|---|
| Based RNN | TNET (*Li et al., 2018*) | CA | ✓ | ✓ | SeqClass |
| | RAM (*Chen et al., 2017*), PBAN (*Gu et al., 2018*) | GA | ✓ | ✓ | Seq2Seq |
| | IAN (*Ma et al., 2017*) | GA | ✓ | X | SeqClass |
| | ATAE-LSTM (*Wang, Huang & Zhao, 2016*), DyMemNN (*Tay, Tuan & Hui, 2017*) | CA | ✓ | X | Seq2Seq |
| Based GCN | KumaGCN (*Chen, Teng & Zhang, 2020*) | CA | ✓ | X | DepenTree |
| | DualGCN (*Li et al., 2021*) | GA | ✓ | ✓ | DepenTree |
| | ASGCN (*Zhang, Li & Song, 2019b*) | DPA | ✓ | ✓ | DepenTree |
| | BiGCN (*Bian et al., 2020*), TGCN (*Tian, Chen & Song, 2021*) | CA | ✓ | ✓ | DepenTree |
| Based PLMs | Post-Training BERT (*Xu et al., 2019*) | DPA | X | ✓ | TokenClass |
| | CG-BERT (*Qiu et al., 2020*) | DPA | ✓ | ✓ | TokenClass |
| | BERT-Pair (*Wu et al., 2020*) | DPA | X | ✓ | TokenClass |
| | RoBERTa (*Dai et al., 2021*) | DPA | X | ✓ | TokenClass |

four Chinese datasets were selected to compare and analyze the three types of models in terms of classification performance, performance with different aspect quantities, specific case performance, and computational cost.

# METHODOLOGICAL DESCRIPTION

## Problem statement

ASC is a fine-grained sentiment analysis task in which sentiment polarity reflects the sentiment orientation of a specific aspect (Aspect), which is usually categorized as positive, negative, or neutral, and the task aims to identify the sentiment polarity of a specified aspect in a sentence. A sentence may contain several different Aspects, each of which may have a different sentiment polarity. Deep learning models are widely used in sentiment classification tasks, many of which are based on a single language (Chinese or English corpus). In this article, we aim to analyze the performance of different deep learning architectures on different languages (English and Chinese) for sentiment polarity classification, hoping to find a model that is suitable for both languages and can achieve good classification results in sentiment analysis. In this section, we first give a specific definition of the ASC task, and then introduce three different deep learning architectures, all of them currently achieving SOTA results in this domain.

For a given set of sentence-aspect pairs $(S, A)$, the n specific aspects corresponding to that sentence are $A = \{a_1, a_2, \ldots, a_n\}$. the specific goal of ASC is to predict the sentiment polarity p, $p \in \mathbb{P} = \{P, O, N\}$, for a specific aspect of a given sentence, $P, O, N$ denote the positive, neutral and negative affective polarities, respectively.

**Table 2  An overview of ASC benchmark datasets.** Where a represents aspect term; c represents aspect category; o represents opinion term; p represents sentiment polarity; the last column indicates the references associated with each dataset.

| Dataset | Language | Annotations | References |
|---|---|---|---|
| SemEval-2014 | English | a,c,p | *Li et al. (2018), Chen et al. (2017), Gu et al. (2018), Ma et al. (2017), Wang, Huang & Zhao (2016), Tay, Tuan & Hui (2017), Chen, Teng & Zhang (2020), Li et al. (2021), Zhang, Li & Song (2019b), Xu et al. (2019), Pontiki et al. (2014), Zheng & Xia (2018), Brun, Popa & Roux (2014), Liu et al. (2018), Li et al. (2019), Tang et al. (2020), Kiritchenko et al. (2014), Majumder et al. (2018), Zeng, Ma & Zhou (2019), Nguyen & Shirai (2015)* |
| SemEval-2015 | English | a,c,p | *Tay, Tuan & Hui (2017), Li et al. (2021), Zhang, Li & Song (2019b), Xu et al. (2019), Pontiki et al. (2015), Tang et al. (2020), Toh & Su (2016), Ma, Peng & Cambria (2018)* |
| SemEval-2016 | English | a,c,p | *Tay, Tuan & Hui (2017), Zhang, Li & Song (2019b), Pontiki et al. (2016), Tang et al. (2020), Toh & Su (2016), Wang et al. (2016), Ruder, Ghaffari & Breslin (2016)* |
| ASC-QA | Chinese | a,c,p | *Wang et al. (2019b)* |
| ASAP | Chinese | c,p | *Bu et al. (2021)* |
| ACOS | English | a,c,p,o | *Cai, Xia & Yu (2021)* |
| ABSA-QUAD | English | a,c,p,o | *Zhang et al. (2021)* |
| Twitter | English | a,c,p | *Li et al. (2018), Li et al. (2021), Zhang, Li & Song (2019b), Bian et al. (2020), Tang et al. (2020), Liu & Zhang (2017), Tang et al. (2016),* |
| Sentihood | English | a,c,p | *Sun, Huang & Qiu (2019), Zhang, Zhang & Vo (2015)* |
| Mitchell | English | a,c,p | *Zhang, Zhang & Vo (2016), Liu & Zhang (2017), Marcheggiani et al. (2014)* |

## Model architecture comparison

Recurrent neural networks play an important role in aspect-level sentiment classification tasks, as shown in Fig. 2A. The input layer takes in a complete sentence, which is processed through word embedding and then fed into the hidden layer. The hidden layer can be a standard RNN, bidirectional RNN, long short-term memory network and so on, attention-based recurrent neural network, and so on.

The output is the specific aspect of the sentence and the sentiment polarity of the target. The graph neural network model used for the ASC task mainly introduces the graph convolutional network (GCN). Its basic architecture involves encoding the target sentence and inputting the encoded sentence into the GCN module, which outputs the representation of the specific target.

By applying attention mechanisms, specific scores can be obtained, which are then inputted into a SoftMax classifier to obtain the final sentiment polarity classification. As for pre-trained language models, the main difference lies in the embedding layer, which uses pre-trained language models such as BERT and its various variants. After pre-training, they are inputted into different deep models to obtain the sentiment polarity of specific aspects.

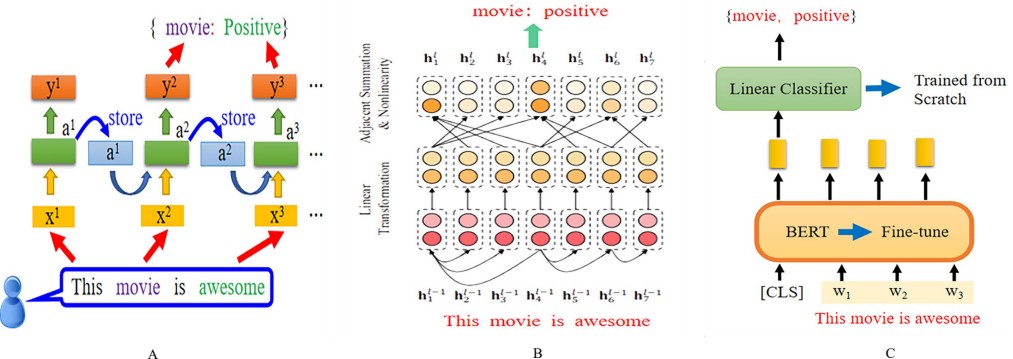
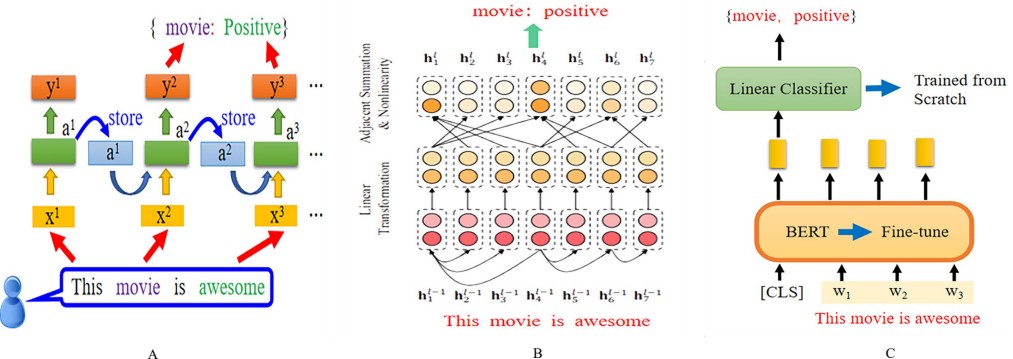

**Figure 2** **Different deep learning models based on ASC structure.** Based RNN, Based GCN (Image quoted from *Zheng & Xia (2018)*), and Based PLMs.

## Model training

(1) RNN-based models

Different deep learning models based on RNN structures can be trained by minimizing the cross-entropy error with a regularization term for the parameters. The loss function is defined as follows:

$$Loss = -\sum_{i \in S}\sum_{a \in A} g_i^a log \hat{g}_i^a + \eta \frac{\|\gamma\|^2}{\gamma \in \mathcal{H}} \tag{1}$$

where $i$ represents the $i$th sentence in the training set, $S$ represents the total number of sentences in the training set, $a$ represents the index of sentiment polarity, $A$ represents the set of sentiment classification, which has two polarities: positive and negative, $g_i$ represents the ground truth distribution of sentiment polarity, which can be the distribution of the target sentence or a one-dimensional one-hot encoded vector (positive sentiment is represented as 1, and negative sentiment is represented as 0), and $\hat{g}_i$ represents the predicted sentiment distribution of the model. $\eta$ is the coefficient of L2 regularization term, $\gamma$ represents the trainable parameters, and $\mathcal{H}$ represents the set of all parameters. The goal of training is to minimize the cross-entropy error $g_i$ with respect to $\hat{g}_i$. To prevent overfitting, the dropout mechanism and early stop mechanism are employed. The backward propagation algorithm using Eq. (2) is used to compute $\gamma$ and update the parameter set $\mathcal{H}$, $\lambda$ represents learning rate.

$$\mathcal{H} = \mathcal{H} - \lambda \frac{\partial Loss(\mathcal{H})}{\partial \mathcal{H}} \tag{2}$$

(2) Models based on GCN

All deep learning models based on GCN have a SoftMax layer at the end to output the probability of sentiment polarity. Its definition is as follows:

$$P = softmax(\mathcal{W}_p r + \mathcal{B}_p) \tag{3}$$

where $r$ is the vector representation of a specific aspect, $P$ is the generated polarity probability distribution, $\mathcal{W}_p$ and $\mathcal{B}_p$ are the parameter vector to be learned. The loss

function is also a standard cross-entropy error, defined as follows:

$$Loss = -\sum_{i=1}^{L}\sum_{a_{i,j}} log P_{g_{i,j}} + \eta \|\gamma\|^2 \tag{4}$$

Where L is the length of the training set, $\gamma$ is the set of parameters to be trained, $\eta$ isthe Penalty coefficient, $a_{i,j}$ is the set of all aspects, $g_{i,j}$ represents the $j$-th training label corresponding to the $i$-th aspect category, and $P_{g_{i,j}}$ represents the probability distribution of aspect category, given by Eq. (3).

(3) Models based on PLMs

All pre-trained language models are based on BERT or its variants, and in ASC tasks, PLMs are used in the sentence embedding part. The embedding result is passed to different deep learning frameworks for training and finally, the sentiment polarity classification result is obtained. Therefore, the definition of the training function is the same as the above two types of models.

## Evaluation metrics

The performance of a model can be evaluated based on its classification effectiveness and commonly used metrics for this purpose include Accuracy, Precision, Recall, F1, and Macro-F1. Accuracy is the most commonly used evaluation metric for classification, and for aspect-based classification since the number of categories is more than two, Accuracy (short for Acc) and Macro-F1 are used as evaluation metrics. For sentiment polarity classification, which has two categories (positive and negative), Accuracy and Macro-F1 are used to measure the classification performance. The definitions of these evaluation metrics are given below: (1) ACCURACY:

$$Accuracy = \frac{TP+TN}{TP+FP+TN+FN} = \frac{TP+TN}{N} \tag{5}$$

(2) F1-MEASURE:

$$F1 = \frac{2TP}{2TP+FP+FN} \tag{6}$$

(3) Macro-F1:

Before giving the definition of Macro-F1, the definitions of Macro-Precision and Macro-Recall are given first. Their calculation formulas are as follows:

$$Macro-Precision = \frac{1}{|C|}\sum_{i=1}^{|C|}\frac{TP_i}{TP_i+FN_i} \tag{7}$$

$$Macro-Recall = \frac{1}{|C|}\sum_{i=1}^{|C|}\frac{TP_i}{TP_i+FP_i} \tag{8}$$

$$Macro-F1 = \frac{2Macro-Precision \times Macro-Recall}{Macro-Precision+Macro-Recall} \tag{9}$$

where $N$ represents the length of the test set, $TP$ and $TN$ represent the number of true positive and true negative predictions, respectively; $FP$ and $FN$ represent the number of false positive and false negative predictions, respectively; and $C$ represents the number of categories.

**Table 3  English datasets.** 'Train' represents the training set and 'Test' represents the test set. For the IMDB dataset, ten-fold cross-validation was used to split the training and testing sets).

| Datasets | Positive(Pos) | | Negitive (Neg) | |
|---|---|---|---|---|
| | **Train** | **Test** | **Train** | **Test** |
| IMDB | 1,000 | | 1,000 | |
| Twitter | 1,561 | 173 | 1,560 | 173 |
| Lap14 | 994 | 341 | 870 | 128 |
| Rest16 | 1,240 | 469 | 439 | 117 |

**Table 4  Chinese datasets.** Ten-fold cross-validation was used to split datasets.

| Datasets | Positive(Pos) | Negitive(Neg) |
|---|---|---|
| Hotel[a] | 5,322 | 2,444 |
| WaiMai[b] | 4,000 | 6,000 |
| SinaWeibo[c] | 5,993 | 5,993 |
| DouBan[d] | 6,000 | 6,000 |

**Notes.**
[a] https://raw.githubusercontent.com/SophonPlus/ChineseNlpCorpus/master/datasets/ChnSentiCorp_htl_all/ChnSentiCorp_htl_all.csv.
[b] https://raw.githubusercontent.com/SophonPlus/ChineseNlpCorpus/master/datasets/waimai_10k/waimai_10k.csv.
[c] https://github.com/SophonPlus/ChineseNlpCorpus/blob/master/datasets/weibo_senti_100k.
[d] https://www.kaggle.com/utmhikari/doubanmovieshortcomments.

# EXPERIMENT

## Experimental dataset

Eight benchmark datasets were used to evaluate the classification performance of different deep learning models, including four English datasets and four Chinese datasets. Each dataset contains two categories of reviews: positive (Pos) and negative (Neg). Specifically, the English datasets include the IMDB dataset, which contains movie reviews with overall sentiment polarity (positive or negative), the Twitter dataset, which consists of individual tweets, and two SemEval datasets: Lap14 and Rest16, which are reviews of computers and restaurants, respectively, each containing aspect-level terms and corresponding polarity. Texts with polarity conflicts and unclear aspect-level terms were removed before model training. Table 3 is the distribution of the four English datasets.

Chinese datasets: The hotel and takeaway datasets are both open-source Chinese review datasets from GitHub, consisting of reviews on hotels and takeaway food, respectively. The Sina Weibo dataset is an open-source Chinese natural language processing corpus consisting of over 100,000 Sina Weibo posts with sentiment labels, with over 50,000 positive and negative comments each. 5,993 comments were randomly selected to construct the Sina Weibo dataset. The DMSC dataset consists of 10,000 comments on Chinese movies from Douban, with 5,000 positive and negative comments each. The specific statistics of these datasets are shown in Table 4. Preprocessing of the Chinese datasets involved: (1) removing all punctuation and special characters from the text, such as meaningless emoticons and various Martian texts, retaining only the more semantically meaningful Chinese text information. (2) There are no spaces between words in Chinese text. The jieba

segmentation tool was used to split sentences containing words such as "but," "however," and "nevertheless" into two sentences. (3) Stop words were removed, but words such as "not," "did not," and "all" that affect sentiment judgments were retained.

## Experimental environment and parameter settings

The experiment was conducted on an Ubuntu 18.04 operating system, using an NVIDIA Tesla V100 GPU, and the development environment was Python 3.6.9 on Google Colab. TensorFlow 1. 15.2 and PyTorch 1. 1.0 were used as learning frameworks for building deep models. For models that provided source code, baseline results were generated on all eight datasets. For models that did not provide source code, the best hyperparameters were set using their reported values in their papers.

To ensure fair comparison, all English datasets were initialized with 300-dimensional Glove embeddings provided by *Pennington, Socher & Manning (2014)*. In addition, because PBAN and BiGCN use positional information, the position nal embedding dimension was uniformly set to 30 for a fair comparison. The spaCy tool was used to obtain dependency relationships. All Chinese datasets were pre-trained with Chinese word embeddings from the Chinese Word Vectors corpus (https://github.com/Embedding/Chinese-Word-Vectors), which was provided by researchers from the Institute of Chinese Information Processing, Beijing Normal University and the DBIIR Laboratory, Renmin University of China. All word embeddings were trained using the ngram2vec toolkit, which is a superset of the word2vec and fast text toolkits that supports abstract context features and models. The dimensionality of all pre-trained Chinese word embeddings was set to 300. For PLMs, pre-trained BERT models were used.

## Results and analysis
### Analysis of classification performance

Tables 5 and 6 show the comparison results of all models in specific aspect classification and sentiment polarity classification. All models were compared on Chinese and English datasets. The table entries in bold with an underline represent the best results. The conclusions drawn from Tables 3 and 4 are as follows:

From Table 3, it can be observed that the PBAN model has the highest accuracy of 84.60% on the IMDB dataset, BiGCN ranks second in accuracy 81.32%. The CG-BERT achieved the highest accuracy of 94.38% on the Twitter dataset, BERT-pair ranks second in accuracy 92.78%; the KumaGCN model had the highest accuracy of 74.59% on the Rest16 dataset, RAM and kumaGCN perform about the same; the KumaGCN model had the highest accuracy of 73.82% on the Res16 dataset, Based on the macro-average F1 score, the BERT-Pair model performed the best on all datasets. Most models had lower performance on Twitter and Lap14 datasets compared to IMDB and Rest16 datasets. The DualGCN and PBAN models had higher macro-average F1 scores on all datasets, indicating their good performance in handling multiple classification tasks.

From Table 4, it can be seen that the CG-BERT model had the highest accuracy of 83.77% in the hotel dataset, and achieved the highest macro-F1score of 77.99% among all models. The RoBERTa model had the highest accuracy of 84. 17% in the takeaway dataset, but its macro-average F1 score was lower at 67.22%. The BERT-Pair model performed

**Table 5  Classification performance of models on English datasets (where AC represents aspect category and SP represents sentiment polarity).** Bold and underlined values indicate the best result.

| Model | AC | | | | SP | | | |
|---|---|---|---|---|---|---|---|---|
| | **IMDB**<br>**Acc. Macro-F1** | **Twitter**<br>**Acc. Macro-F1** | **Lap14**<br>**Acc.Macro-F1** | **Rest16**<br>**Acc. Macro-F1** | **IMDB**<br>**Acc. F1** | **Twitter**<br>**Acc. F1** | **Lap14**<br>**Acc. F1** | **Rest16**<br>**Acc. F1** |
| ATAE-LSTM | 69.21 68.24 | 69.65 67.40 | 69.14 63.18 | 83.25 63.85 | 77.20 68.70 | 76.60 68.90 | 70.80 69.00 | 71.50 69.50 |
| RAM | 70.23 70.80 | 69.36 67.30 | 74.49 71.35 | **85.58** 65.76 | 70.80 80.23 | 74.49 **71.35** | 69.36 73.85 | 72.14 60.24 |
| IAN | 73.72 69.21 | 60.49 67.33 | 74.31 58.28 | 85.44 65.99 | 78.60 72.10 | 76.70 68.29 | 76.70 **76.70** | 72.31 65.35 |
| PBAN | **84.60 76.60** | **78.74 72.61** | **87.81** 74.12 | 81.67 73.68 | **81.16 74.12** | **78.75** 70.51 | **78.66** 70.84 | **78.04** 70.74 |
| BiGCN | 81.32 71.72 | 72.20 70.45 | **75.63** 70.74 | 87.71 67.87 | 82.93 75.29 | 76.27 72.39 | 75.04 73.85 | 74.59 72.86 |
| ASGCN | 80.77 72.02 | 72.15 70.40 | 75.55 71.05 | **88.99** 67.48 | 71.92 70.63 | 73.51 68.83 | 79.40 69.43 | 79.40 61.18 |
| DualGCN | **83.10** 73.01 | 73.29 72.02 | 75.53 **72.01** | 86.24 67.62 | 84.27 **78.08** | 78.84 **74.74** | 75.92 **74.29** | 74.45 69.37 |
| KumaGCN | 81.97 **73.38** | **74.16 73.35** | 74.59 71.84 | 88.96 **70.84** | **89.39** 73.19 | **89.12** 70.89 | **89.23** 72.04 | **88.64 73.82** |
| PT-BERT | 80.98 72.04 | 73.82 69.35 | 72.10 69.90 | 81.16 73.20 | 73.98 69.94 | 72.11 70.74 | 81.33 73.57 | 74.96 70.93 |
| BERT-Pair | 87.9 **79.8** | 92.78 89.07 | 93.30 90.89 | 86.90 83.70 | 97.00 **93.70** | 93.57 **90.83** | 95.60 **92.18** | 89.90 85.90 |
| RoBERTa | 81.42 71.79 | 75.36 71.11 | 73.78 72.37 | 82.76 75.25 | 77.43 74.21 | 75.43 74.04 | 81.60 72.48 | 75.96 71.96 |
| CG-BERT | **89.10** 79.70 | **94.38 90.97** | **95.60 92.64** | **90.10 86.80** | **97.20** 93.60 | **94.27** 90.12 | **95.80** 92.14 | **90.40 86.90** |

Cao and Huang (2023), *PeerJ Comput. Sci.*, DOI 10.7717/peerj-cs.1578

Cao and Huang (2023), *PeerJ Comput. Sci.*, DOI 10.7717/peerj-cs.1578

**Table 6 Classification Performance of Models - Chinese Dataset.** AC represents aspect category, and SP represents sentiment polarity. Bold and underlined values indicate the best result.

| Model | AC Hotel Acc. Macro-F1 | AC WaiMai Acc. Macro-F1 | AC SinaWeibo Acc. Macro-F1 | AC DouBan Acc. Macro-F1 | SP Hotel Acc. F1 | SP WaiMai Acc. F1 | SP SinaWeibo Acc. F1 | SP DouBan Acc. F1 |
|---|---|---|---|---|---|---|---|---|
| ATAE-LSTM | **76.69 67.93** | 62.74 **63.72** | 70.06 70.06 | 64.53 51.12 | 77.63 69.79 | 65.66 38.51 | 67.40 65.16 | 62.18 58.47 |
| RAM | 67.55 62.25 | 60.78 59.73 | 67.55 67.55 | 67.55 49.09 | 60.44 77.23 | 63.28 32.54 | **86.51** 55.29 | 81.06 42.31 |
| IAN | 68.50 64.11 | 62.69 60.90 | 68.50 68.50 | 68.50 **57.10** | 63.41 **77.78** | 71.09 30.77 | 86.22 **59.48** | **81.78** 41.43 |
| PBAN | 70.06 66.14 | **63.05** 62.94 | **76.79 76.79** | **70.06** 50.98 | **82.25** 66.84 | **88.56 60.94** | 82.74 50.57 | 77.68 **69.01** |
| BiGCN | 75.18 68.45 | 60.25 58.44 | 75.18 75.18 | 75.27 66.30 | 77.23 67.54 | 78.75 70.51 | **77.14** 67.61 | **78.30 69.11** |
| ASGCN | **78.66** 69.01 | 66.93 63.47 | 70.84 66.46 | 67.74 64.31 | **77.77** 69.51 | **68.81 63.41** | 68.50 62.66 | 67.08 62.02 |
| DualGCN | 76.34 65.95 | 75.98 67.01 | 76.70 **68.29** | 76.25 68.71 | 75.18 68.45 | 77.41 68.58 | 73.39 62.74 | 77.41 68.38 |
| KumaGCN | 77.69 **70.84** | 77.41 69.06 | 78.04 61.64 | **78.30 70.74** | 69.44 **69.61** | **78.84 72.84** | 76.79 **68.68** | 76.79 67.93 |
| PT-BERT | 72.31 65.35 | 79.96 77.99 | 82.55 76.81 | 82.18 **77.28** | 82.63 78.34 | 83.71 74.44 | 80.16 80.00 | **90.29** 79.41 |
| BERT-Pair | 81.43 69.36 | 81.64 76.33 | 81.48 76.21 | 81.67 76.21 | **88.96 80.56** | 83.80 **80.79** | 81.31 79.41 | 77.49 75.50 |
| RoBERTa | 82.54 70.75 | **84.17** 67.22 | 82.07 70.10 | 81.16 72.33 | 81.87 72.89 | 82.31 73.53 | 76.33 72.76 | 73.84 72.66 |
| CG-BERT | **83.77 77.99** | 82.97 **79.88** | **84.19 78.58** | **83.25** 75.53 | 83.11 78.70 | **89.01** 78.34 | **90.02 80.79** | 84.05 **83.12** |

the best in macro-average F1 score, reaching 80.56%. The PT-BERT model performed best in the Sina Weibo dataset, with both accuracy and macro-average F1 score exceeding 80%. Both PT-BERT and CG-BERT models performed well in the Douban dataset, with the PT-BERT model achieving the highest macro-average F1 score of 90.29%. Overall, pre-trained models such as PT-BERT, RoBERTa, and CG-BERT performed well in these sentiment analysis tasks, but their performance varied across different tasks. In addition, graph neural network models such as BiGCN, ASGCN, DualGCN, and KumaGCN also performed well on some tasks, but not on others. Therefore, it is necessary to choose asuitable model for specific tasks.

### Performance on aspect quantity

Regardless of whether it is an English or Chinese dataset, a sentence may contain multiple aspects, so the influence of aspect quantity on the model is also worth noting. We divided all Chinese and English training datasets into different groups according to the different numbers of aspects, and calculated the classification accuracy among these groups. For comparison purposes, we removed sentences with more than seven aspects because the sample size for these sentences is too small. We selected the Twitter and Lap14 English datasets, as well as the Eleme and Douban movie review Chinese datasets for comparison. According to the analysis results of Tables 3 and 4, PBAN, KumaGCN, and CG-BERT have the highest average accuracy on both Chinese and English datasets. Therefore, we selected these three models to analyze their performance on different aspect quantities.

The visualization results are shown in Fig. 3. The PBAN model(a) performed similarly on the Twitter, lap14, and Douban datasets, with accuracy decreasing as the number of aspects increased.The KumaGCN model(b) performed best on the Twitter dataset, followed by the Lap14 dataset, however, as the number of aspects increased, the overall classification accuracy of the model decreased. The classification performance of the CG-BERT model (c)on lap14 is unstable, and when the number of aspects increases, the classification accuracy tends to go to market, and the performance is best on the Douban dataset on average, and the performance is relatively stable on the other two datasets. Overall, these models generally decrease in classification accuracy with the increase in the number of aspects, about 60% to 70%, and when the number of aspects is small, the classification accuracy of the model is also low, and their performance on Chinese datasets is inferior to the English dataset.

### Specific case analysis

We are also concerned about the importance of terminology in specific aspects of different models.The visualization results are shown in Fig. 4, where darker colors indicate more important words. "X" represents model prediction errors, and " $\sqrt{}$" represents model prediction correctness. For the first sentence in (a), "The lamb kebab was okay, but not very fresh," for the "lamb" aspect, none of the three models can focus well on the adjacent words and lexical structure of the aspect term, and put more weight on positive words like "okay" and "fresh," giving less weight to negative words, resulting in prediction errors. For the "room" aspect in the second sentence(b), only PBAN predicts incorrectly, as it focuses more on the positive term "big" but ignores the position information and sentence

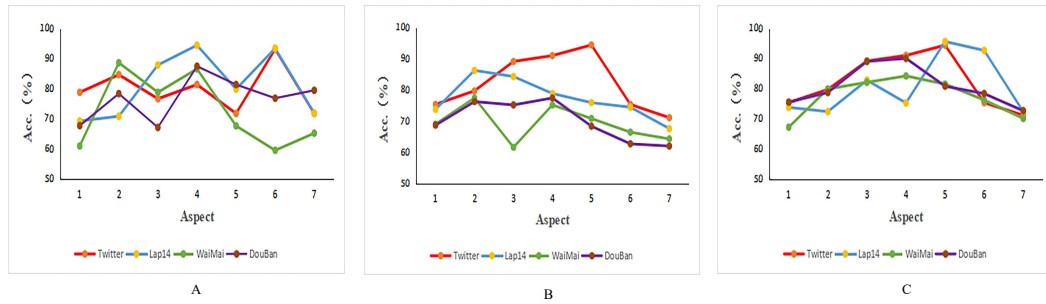

**Figure 3** **Accuracy of models on different numbers of aspects.** (A) PBAN. (B) KumaGCN. (C) CG-BERT.

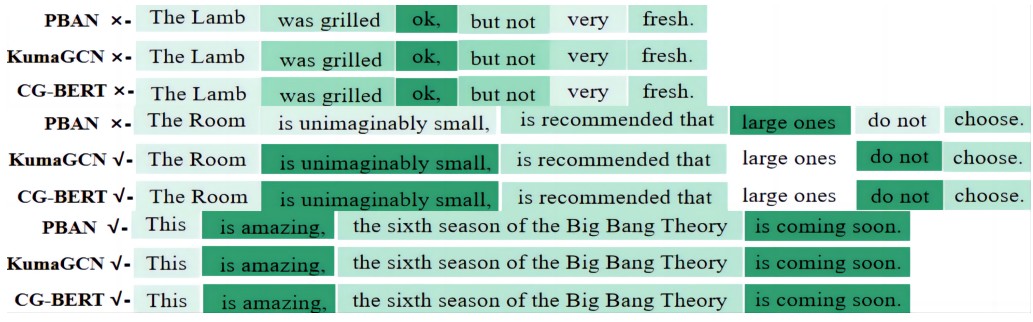

**Figure 4** **The importance of visualization weight for aspect terms.** Aspect: the lamb; Polarity: negative. Aspect: the room; Polarity: negative and Aspect: The Big Bang Theory; Polarity: positive.

syntax, resulting in prediction errors. The other two models can predict well, as KumaGCN and CG-BERT pay attention to negation words such as "small," "cannot," and "don't," and also consider their positions and grammatical structures, linking "suggestion" and "don't" together to make correct judgments. For the "The Big Bang Theory" aspect in the third sentence(c), all three models give a high weight to positive words like "amazing" and "is coming soon," resulting in correct predictions. Therefore, these three types of models cannot make accurate judgments on Chinese contrastive sentences.

### Time complexity analysis

In addition to analyzing the classification performance of deep learning models, the analysis of the time complexity of the models also reflects their performance. This section analyzes the computational cost of all models, and the specific results can be found in Table 7, with time units in seconds.

Since the number of texts in the Chinese dataset is larger than that in the English dataset, the training time is relatively longer. Twitter is the largest English dataset, and the training time is correspondingly increased compared to other English datasets. However, a separate comparison shows that models with graph neural networks have lower computational costs than models with attention mechanisms. DualGCN has the shortest training time on the IMDB dataset, while PT-BERT has the shortest training time on the Twitter dataset.

**Table 7  Comparison of training time for all models.** Bold and underlined values indicate the best result.

| Model | IMDB | Twitter | Lap14 | Rest16 | Hotel | WaiMai | SinaWeibo | DouBan |
|---|---|---|---|---|---|---|---|---|
| ATAE-LSTM | 2,662 | 5,185 | **2,855** | **2,940** | 8,185 | 9,952 | 8,136 | 7,893 |
| RAM | 3,855 | 5,915 | 4,065 | 4,560 | 8,915 | 9,083 | 10,775 | 8,624 |
| IAN | 4,638 | 6,672 | 4,738 | 4,560 | 8,672 | 9,662 | 8,408 | 9,758 |
| PBAN | 4,838 | 6,952 | 4,080 | 5,580 | 8,952 | 10,046 | **7,462** | 9,103 |
| BiGCN | 2,843 | 3,731 | 3,726 | 4,004 | **5,838** | 5,679 | 8,246 | **6,389** |
| ASGCN | 2,440 | 3,626 | 3,120 | **3,565** | 7,210 | 8,169 | **7,775** | **7,462** |
| DualGCN | **2,254** | 3,510 | **2,998** | 3,728 | 9,125 | 8,765 | 9,840 | 8,875 |
| KumaGCN | **2,775** | 3,854 | 4,125 | 4,776 | 8,340 | 8,581 | 9,613 | 8,310 |
| PT-BERT | 3,438 | **4,317** | 4,267 | 6,040 | **5,388** | **6,796** | 8,408 | 11,389 |
| BERT-Pair | 3,404 | 9,266 | 4,201 | 4,655 | 8,102 | 7,691 | 10,340 | 11,462 |
| RoBERTa | 3,542 | 9,105 | 3,989 | 4,287 | 12,501 | 9,657 | 10,000 | 12,475 |
| CG-BERT | 3,757 | **4,548** | 5,251 | 5,767 | 9,403 | 8,195 | 10,083 | 10,760 |

Among the four Chinese datasets, PBAN has the lowest computational cost on the Sina Weibo dataset, and BiGCN has the lowest computational cost on the food delivery and Douban movie review datasets. Overall, PLMs have higher time complexity, especially on the Chinese dataset, and all models have an average training time on the English dataset lower than on the Chinese dataset.

## CONCLUSION

In this study, we evaluated different deep learning models for aspect-level sentiment analysis on both English and Chinese datasets. Our results showed that PBAN, KumaGCN, and models using CG-BERT had better predictive performance compared to other models on both English and Chinese datasets. While PBAN was slightly inferior in terms of computational cost, KumaGCN performed well in all aspects. Pre-trained language models showed better classification performance on Chinese datasets compared to the other two types of models, but overall, all models performed worse on Chinese datasets than on English datasets. Our work established a reference baseline for cross-lingual sentiment analysis tasks and evaluated the strengths and weaknesses of different deep learning models on different data distributions and word embedding, contributing to the establishment of multi-lingual aspect-level sentiment classification tasks in the future. We used the Glove method for English word embedding representation and the ngram2vec method for Chinese word embedding representation, which may have affected the classification performance of the models on these two types of corpora. Future work could explore the possibility of pre-training a unified classification model that can achieve good classification performance with minimal fine-tuning on any sentiment classification dataset. It should be noted that the performance of deep learning models on aspect-level sentiment classification tasks also depends on factors such as data quality, data volume, and hyper parameter settings. Therefore, in practical applications, these factors need to be considered comprehensively when selecting an appropriate model and conducting optimization.

Although there is not much difference between Chinese and English sentiment analysis in terms of algorithm, their syntactic structures are different: Chinese requires participles; English requires word reductions (tense, singular and plural, *etc*). In addition, Chinese sentiment analysis also faces problems such as insufficient dictionary resources and difficulty in distinguishing sentences that require association, Solving these problems requires a combination of tools to improve the accuracy of sentiment analysis, this article does not give solutions, but only analyzes and reviews the performance of various models. We hope to come up with effective solutions for cross-language migration. Another area of concern is the development of quality Chinese dictionary resources in the future.

### Funding
The authors received no funding for this work.

### Competing Interests
The authors declare there are no competing interests.

### Author Contributions
- Feifei Cao conceived and designed the experiments, performed the experiments, analyzed the data, performed the computation work, prepared figures and/or tables, authored or reviewed drafts of the article, and approved the final draft.
- Xiaomin Huang analyzed the data, authored or reviewed drafts of the article, and approved the final draft.

### Data Availability
The data are available at:

Hotel: https://raw.githubusercontent.com/SophonPlus/ChineseNlpCorpus/master/datasets/ChnSentiCorp_htl_all/ChnSentiCorp_htl_all.csv.

- WaiMai: https://raw.githubusercontent.com/SophonPlus/ChineseNlpCorpus/master/datasets/waimai_10k/waimai_10k.csv.

- SinaWeibo: https://github.com/SophonPlus/ChineseNlpCorpus/blob/master/datasets/weibo_senti_100k.

- DouBan: https://www.kaggle.com/utmhikari/doubanmovieshortcomments.

jinhuakst, IAMZn, & iamzn. (2023). caofeifei946198/ChineseNlpCorpus: Sentiment-Analysis (Version 946198). Zenodo. https://doi.org/10.5281/zenodo.8289105.

- IMDB Sentiment Analysis (stanford.edu): https://ai.stanford.edu/~amaas/data/sentiment/.

- Li Dong, Furu Wei, Chuanqi Tan, Duyu Tang, Ming Zhou, and Ke Xu. 2014. Adaptive Recursive Neural Network for Target-dependent Twitter Sentiment Classification. In Proceedings of the 52nd Annual Meeting of the Association for Computational Linguistics (Volume 2: Short Papers), pages 49–54, Baltimore, Maryland. Association for Computational Linguistics.

- Maria Pontiki, Dimitris Galanis, John Pavlopoulos, Harris Papageorgiou, Ion Androutsopoulos, and Suresh Manandhar. 2014. SemEval-2014 Task 4: Aspect Based Sentiment Analysis. In Proceedings of the 8th International Workshop on Semantic Evaluation (SemEval 2014), pages 27–35, Dublin, Ireland. Association for Computational Linguistics.

- Maria Pontiki, Dimitris Galanis, Haris Papageorgiou, Ion Androutsopoulos, Suresh Manandhar, Mohammad AL-Smadi, Mahmoud Al-Ayyoub, Yanyan Zhao, Bing Qin, Orphée De Clercq, Véronique Hoste, Marianna Apidianaki, Xavier Tannier, Natalia Loukachevitch, Evgeniy Kotelnikov, Nuria Bel, Salud María Jiménez-Zafra, and Gülşen Eryiğit. 2016. SemEval-2016 Task 5: Aspect Based Sentiment Analysis. In Proceedings of the 10th International Workshop on Semantic Evaluation (SemEval-2016), pages 19–30, San Diego, California. Association for Computational Linguistics.

## Supplemental Information

Supplemental information for this article can be found online at http://dx.doi.org/10.7717/peerj-cs.1578#supplemental-information.

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
