# Peer review of "Performance analysis of aspect-level sentiment classification task based on different deep learning models"

_PeerJ Computer Science, doi:10.7717/peerj-cs.1578_

## Round 0.1 · original submission · Major Revisions

Please address the reviewers' comments.

Reviewer 1 ·

Basic reporting

The language of the text is ok with only minor errors.
Literature references are ok but it would be useful to add some positions especially from authors outside China to give more international background of the paper.
The article structure is ok.
The formal results contains all material needed.

Experimental design

The topic of the paper is within Aims and Scope of the Journal.
The paper lacks explicitly described research gap, the goal of the paper and research questions.
Authors should add them and link to literature.
On the basis of what literature analysis they specify the research gap. On the basis of what literature positions and analysis the questions were formulated. How research goal and research questions are related to the research gap.
The description of the research methods are ok, but I think the problem statement should be described in more in-deep way.

Validity of the findings

The paper lack discussion with other literature – Authors should add it as a whole section to the paper.
The limitations of the paper should be described.
Please write what is new in your paper how it differs from others similar papers existing in per reviewed journals.

Reviewer 2 ·

Basic reporting

The paper focuses on a narrow but important topic. At first glance, the research seems to be suitable for publishing as it is, but after careful consideration, there are some minor changes to e made in order to fit the high expectations of the journal.
The title is appropriate, covers and reflects the topic and attractive too. The abstract is compact, well-written and understandable. The introduction highlights the context and research goals, with an appropriate literature background.
The literature review is prepared accordingly, and the relevant and important international sources are analyzed critically and analytically. The methodology is described accordingly and detailed, it helps a lot to understand the importance and logic of the results. The results are relevant and supported by the methodological toolset.
My recommendations for improving the paper:
- row 120: the sentence starts with "3." - it should be a mistyping;
- 2nd chapter: instead of "Related work" I would write "Literature review"
- 3. Start: this is the methodological description, it is better to use this term (Methodology)
- Limitations of the research should be better highlighted, preferably in the last chapter
- Implications of the research are not described.

Experimental design

The research can be regarded as original due to its topic, unique methodological approach, and gap-filling results.

Validity of the findings

The findings are appropriate, and well-supported by the methodology and the results/discussions.

Additional comments

n/a

---

## Round 0.2 · accepted · Accept

The previous Academic Editor is not available and so I have taken over handling this submission. I would like to thank the authors for carefully addressing the reviewers' concerns from the previous round and for providing a detailed response letter.

Based on the review of one of the original reviewers as well as my own, I'm happy to recommend this paper for acceptance in its current form. Well done, congratulations!

Reviewer 1 ·

Basic reporting

The Authors implemented my remarks.

Experimental design

The Authors implemented my remarks.

Validity of the findings

The Authors implemented my remarks.